# Concepts of psychosocial distress and help-seeking preferences among Indigenous adolescents: A qualitative study from Jharkhand, India

Subhashree Samal[1], S. Prathyush[1], Sumitra Gagrai[1], Nibha Das[1], Nitima Lamay[1], Savitri Banra[1], Suresh Purty[1], Madhuri Purty[1], Birsingh Mundri[1], Dumbi Bodra[1], Mati Murmu[1], Sunita Kalundia[1], Yadav Leyangi[1], Nirmala Nair[1], Prasanta Tripathy[1], Patrick Smith[2], Andrea Danese[2], Kelly Rose-Clarke[3], Abhijit Nadkarni[4,5], Urvita Bhatia[4,5], Saicha Naik Bandiwadekar[4], Brinda Singh Raikwar[4], Devika Gupta[4], Sachin Barbde[1‡], Audrey Prost[3‡*]

**1** Ekjut, Chakradharpur, Jharkhand, India, **2** Institute of Psychiatry, Psychology and Neuroscience, King's College London, London, United Kingdom, **3** Institute for Global Health, University College London, London, United Kingdom, **4** Addictions and Related Research Group, Sangath, Goa, India, **5** Centre for Global Mental Health, Department of Population Health, London School of Hygiene and Tropical Medicine, London, United Kingdom

‡ Sachin Barbde and Audrey Prost are joint senior authors.
* Audrey.prost@ucl.ac.uk

## Abstract

Mental health disorders affect around one in seven adolescents globally. In India, home to one-fifth of the world's adolescent population, attention to adolescent mental health is increasing, but access to care remains limited, particularly for Adivasi (Indigenous) adolescents. Understanding how Adivasi adolescents conceptualise psychosocial distress and the kinds of support they value is essential to design supportive interventions. We conducted a qualitative study in rural West Singhbhum, Jharkhand. Using purposive sampling, we recruited 88 participants: 53 adolescents aged 10–19 years, 13 teachers, 11 parents, three community health workers, and eight mental health programme staff aged 18–25 years. Trained peer interviewers conducted semi-structured interviews and group discussions with adolescents in the Ho Adivasi language. A team including experienced Ho researchers and peer interviewers analysed data using the Framework approach. Adolescents, parents, and teachers used variations of the Ho root word "udu" to describe distress. "Udu" denoted thoughts or worries, depending on context. Sources of "udu" included family responsibilities and societal expectations that grew with age: girls were expected to do household work and received less support for school, while boys described income-earning pressures and a lack of emotional outlets. Other "udu" causes included violence at home, school, and public spaces, as well as family separation through migration, remarriage, or parental loss. Impacts of severe "udu" ranged from social withdrawal and strained relationships to early marriage, migration for work, and suicide attempts.

**Data availability statement:** In compliance with our ethical approval from the Independent Ethics Committee linked to Ekjut and from University College London's Research Ethics Committee (ID:1881/007), individual transcripts from this study cannot be publicly shared. Participants did not consent to the disclosure of these transcripts, and they contain sensitive and potentially identifying information. Therefore, public access would compromise participant confidentiality. All relevant data are included within the manuscript and S1 Data.

**Funding:** The research was funded by a grant from the UKRI Medical Research Council (Reference: MR/T040419/1). The following authors received salary support from this grant: SS, PS, SG, ND, NL, SB, SP, MP, BM, DB, MM, SK, YL, PS, AD, KR-C, AN, UB, SNB, BSR, DG, SB, AP. The funders had no role in study design, data collection and analysis, decision to publish, or preparation of the manuscript.

**Competing interests:** The authors have declared that no competing interests exist.

Adolescents experiencing "udu" preferred informal support from friends, siblings, or teachers, while those with experience of help from respectful community health workers or trusted, trained older peers valued these. Group-based support was widely acceptable. Strengthening access to early care, alongside measures supporting family livelihoods and social protection, could improve Ho and broader Adivasi adolescent mental health.

## Introduction

Mental disorders affect nearly one in seven adolescents aged 10–19 years globally, with anxiety, depression, and behavioural disorders the most common [1,2]. Left unaddressed, these disorders can disrupt overall well-being, education, relationships, often continuing into adulthood with wide-ranging consequences [3–5].

India is home to one in five of the world's adolescents [6]. According to the 2015–16 National Mental Health Survey, 7% of those aged 13–17 years experience mental health problems [7]. While the treatment gap for mental health disorders in adolescents in India is largely unknown, it is likely to exceed the already high 85% treatment gap reported for adults [7].

Evidence on adolescent mental health from Indian rural and Adivasi (Indigenous) communities is particularly scarce. Adivasi communities – who constitute around 9% of India's population – are highly heterogenous, but likely to be disproportionately affected by poor mental health due to pre-existing socio-economic, environmental, and health vulnerabilities [8]. Only a handful of studies have focused specifically on Adivasi adolescents. One community-based cross-sectional survey of 623 Adivasi adolescents aged 11–19 years in Gujarat, Tamil Nadu and Meghalaya found a 16% prevalence of mental disorders, more than double the national estimate of 7% from the National Mental Health Survey [9]. Another survey, conducted in 2016–17 with 3,324 adolescent girls aged 10–19 years in Adivasi blocks of rural Jharkhand, found that 12% had symptoms of internalising or externalising problems [10]. Together, these limited data suggest that Adivasi adolescents face at least as great, if not greater, mental health burdens than their peers in the general population.

Beyond prevalence however, little is known about how Adivasi adolescents and adults understand common mental disorders, how they experience and express them, or the forms of support they value [11]. Psychosocial distress - both a risk factor for, and a symptom of, common mental disorders - is a useful entry point for understanding common mental disorders in community contexts where diagnostic categories such as depression and anxiety are seldom used [12]. Adolescents' concepts of distress are likely to shape whether, when, and from they seek help, so understanding them is essential to develop effective interventions [13].

Social context further underscores this need. The most recent national assessment (2011–12) found that 45% of rural Adivasi families lived below the poverty line [14]. In addition, ethnographic studies from Northern and Eastern India show that

Adivasi adolescents often navigate tensions between their Indigenous identity and aspirations for social recognition and economic mobility. These pressures can generate conflict within individuals and families [15,16]. Such complex social dynamics underscore the need for a grounded understanding of how Adivasi adolescents experience distress.

Our study aimed to explore Adivasi adolescents' concepts of psychosocial distress and their help-seeking preferences in rural Jharkhand. Specifically, we sought to understand local terms used by adolescents, caregivers, teachers and community health workers to describe distress, adolescents' perspectives on its causes and impacts, and the forms of help they considered valuable.

## Methods

### Ethics statement

The study received ethical approval from a government-registered Institutional Ethics Committee linked to Ekjut and from University College London's Research Ethics Committee (Reference 1881/007). We sought individual written informed consent by signature or thumbprint from all participants. Married adolescents or those aged 18 years and over directly provided consent. For unmarried adolescents aged 10–17 years, we sought consent from caregivers and assent from the participants. Two adolescents who were approached did not want to take part because they felt shy. As per our distress protocol, two adolescents with severe distress were followed up by counsellors who spoke with their parents, and one was referred to a clinic-based senior counsellor. Conducting the interviews and reflecting on them during analysis was emotionally challenging for peer researchers. Several had experienced distress in the past and described the process as "revisiting their own lives and troubles". The study team supported peer researchers through repeated, careful debriefing conversations.

### Study design and setting

We did a qualitative study with semi-structured interviews and group discussions in a rural block of West Singhbhum district in Jharkhand, eastern India. West Singhbhum has a predominantly rural, hilly, forested terrain; 64% of residents are Adivasi, most of whom are from the Ho community [17]. The primary sources of livelihood for West Singhbhum families are rainfed agriculture, seasonal migration for work, and daily wage labour. At the last National Family Health Survey (2019–2021), an estimated 77% of men and 56% of women in the district could read [18].

Access to mental health services in West Singhbhum remains limited despite national and state-level policy frameworks [19]. Although the District Mental Health Programme (DMHP) has been approved for all 24 districts in Jharkhand, it is operational in only nine, and mental health services at Health and Wellness Centres are still nascent [20]. Similarly, while the National Adolescent Health Programme (*Rashtriya Kishore Swasthya Karyakram*) includes provisions for adolescent mental health counselling, these services currently focus primarily on sexual and reproductive health [21,22].

The study was led by the civil society organisation Ekjut (https://ekjutindia.org) and embedded within its community mental health programme as part of formative research to develop or adapt an early intervention for adolescent anxiety and depression. Ekjut has worked in West Singhbhum since 2002 to improve the health of rural, Indigenous communities. Its participatory learning and action approach with women's groups has contributed to large reductions in neonatal mortality and maternal depression in Jharkhand [23]. In 2018, Ekjut launched a dedicated mental health programme to provide community-based rehabilitation for adults and adolescents with severe psychosocial disabilities, including schizophrenia, bipolar disorder, severe depression, and substance use disorders. The programme includes a collaborative clinic with the Central Institute of Psychiatry in Ranchi, door-to-door counselling and home visits, peer support group meetings, a social contact initiative to reduce stigma, and referrals to local services offering livelihood support and social protection for people with psychosocial disabilities and their families.

PLOS Mental Health

## Overall methodological orientation

Our overall methodological orientation was shaped by traditions of research with Indigenous communities, including relational accountability and Two-Eyed Seeing [24,25]. We were inspired by Tynan's view that "rather than researchers being accountable to universities and grant funders, first and foremost, relational accountability teaches that researchers are accountable to their relations which includes family, Country and their research communities" [24, p.147]. This shaped our decision to centre the knowledge of local Indigenous researchers of different generations. We also drew on the concept of Two-Eyed Seeing, as introduced by Mi'kmaw Elder Albert Marshall from Unama'ki–Cape Breton. This practice encourages seeing through both Indigenous and non-Indigenous knowledge systems simultaneously and using both for the benefit of all [25]. In our work, this meant centring Adivasi experiences of distress, including local terms in Ho language and existing forms of support, while also engaging with biomedical and psychological frameworks where relevant. For example, although some expressions of distress extended beyond those typically identified through biomedical screening tools and interventions, others such as *low mood*, were also likely to resonate locally given their well-documented cross-cultural occurrence. Our interest in Two-Eyed Seeing also meant creating a space for using indigenous language during data collection, analysis and in this publication, rather than relying solely on translated and highly synthesised content more likely to 'erase' local ways of knowing and doing.

## Participants

We sought to purposively sample participants based on age (10–14 or 15–19 years), sex (with a balance of male and female participants), and marital status (as married adolescents are likely to experience specific stressors). For semi-structured interviews, we specifically sought adolescents with experience of stressors such as living with a single parent, parental migration or neglect, financial hardship at home, or being out of school. For group discussions, we sought views from all willing adolescents in interviewers' own villages, as described below. In addition to adolescents, we also spoke with caregivers (including parents, uncles and aunts) and community health workers including Accredited Social Health Activists (incentivised community health volunteers performing a range of health promotion, prevention, and referral activities under India's National Health Mission), Anganwadi Workers (frontline salaried workers who play a key role in improving child health, nutrition, and early development), and Auxiliary Nurse Midwives (salaried health workers who support a range of primary care activities). Adolescents with noticeable symptoms of severe mental disorders or neurodevelopmental disorders were excluded.

## Data collection

Data were collected by four trained peer interviewers (two men and two women) aged 18–25 years (BM, DB, MM, MG). We worked with peer interviewers because they were closer in age to participants than the rest of the study team, had experience of interacting with adolescents in the local area, and spoke Ho, a language widely understood by adolescents locally. The lead researcher has an MPhil in Women's Studies, and all peer interviewers had completed 12 years of schooling and were bilingual in Hindi and Ho. Data collection tools were initially developed in English and then translated into Hindi as some community health workers and teachers mainly spoke Hindi - and Ho. These tools, including topic guides, were iteratively adapted and refined based on pilot testing and interviewer feedback as part of our attempt at Two-Eyed Seeing. We did not use terms like 'mental illness' or 'common mental disorders' in topic guides and during discussions to avoid imposing external categories onto adolescents' experiences. Instead, the guides used relatable vignettes, like a story about an adolescent girl in the community and probes for describing someone "feeling sad, thinking or worrying too much in a way that is different than normal". All topic guides in Hindi and English are available in S1 Text.

Peer interviewers conducted semi-structured interviews and group discussions in their own villages. They first discussed sources of distress together with the Ekjut team, then used their own social networks to identify supportive

teachers from schools within their "panchayat" (administrative grouping of c.5 villages) and adolescents who had experienced distress due to potentially challenging experiences, as described above in the section on participants in semi-structured interviews. All interviews were face-to-face.

We modified our data collection approach as the study progressed to maximise engagement across groups. For example, initial focus group discussions with young adolescents (6–12 participants) suggested that they were shy and reluctant to talk about emotions and mental health in large group settings. In addition, organising focus group discussions was challenging as adolescents were often involved in household chores and lived in sparsely populated areas, making it challenging to gather larger groups. We therefore changed the approach to holding small, single sex group discussions with 4–5 adolescents, especially for younger adolescents. A detailed breakdown of data collection methods by participant type is in S1 Table.

All interviews and group discussions were conducted in quiet, secluded community spaces away from earshot, audio recorded and lasted between 45 and 65 minutes. We used 'a priori thematic saturation', i.e., the degree to which identified codes or themes are exemplified in the data based on our major themes derived from our broad study objectives, but also 'inductive thematic saturation' based on comparing and contrasting data from within respondent types. For example, we examined whether new sub-themes emerged from interviews or group discussions with younger v. older adolescents and checked whether sub-themes and codes within themes had major gaps [26].

## Analysis

Our qualitative data analysis team included senior Ekjut team members belonging to the Ho community with extensive experience in qualitative research and deep contextual understanding, as well as Ho-speaking peer interviewers (SG, SP, MM, BM, MP) and an early career researcher (SS). Audio recordings of interviews and group discussions were translated from Ho to Hindi by SG, SP, MM, BM. Data were analysed using the Framework approach by SS, SG, NL, ND, SK, MM, SBanra, SBarbde, using manual coding on handwritten transcripts and then summarising themes and indexing in Excel [27]. The analysis followed five key steps. First, the whole team familiarised themselves with the data by reviewing transcripts and notes to identify key themes. Next, they developed a thematic framework based on the study objectives and emerging insights from their initial readings. Major themes reflected the study objectives and sub-themes reflected inductive coding. For example, 'impacts of distress' was a major theme in our topic guide and remained so in our analysis, with types of impacts coded as sub-themes. SS then applied this framework to the data through a process of indexing, where transcripts and observation notes were annotated. In the charting phase, SS rearranged and summarised the data into a matrix by case and according to the thematic framework, checking understanding and resolving coding uncertainties with Ho-speaking team members. The final version of this matrix is available in S1 Data. Finally, during mapping and interpretation, the team compared results across cases, then discussed relationships between themes and overall interpretation. In the results section on concepts of distress, we provide Ho terms with English translations; in subsequent sections, we provide the English translation to keep reporting succinct.

## Positionality statement

The researchers' backgrounds played a crucial role in interpreting concepts of distress described in the data. The early career researcher who led the qualitative study (SS) has worked with local Adivasi communities for nine years through Ekjut, with a particular focus on adolescent health and mental health. However, she is non-Adivasi and does not speak Ho. This may have led to misinterpreting some findings, though SS also had the benefit of being a 'partial outsider', allowing her to check and reflect on understanding with native Ho speakers [28]. Other Ekjut-based researchers, including peer researchers, were from nearby rural Adivasi communities. As local youth, the peer interviewers shared a strong connection with adolescent participants, along with a deep understanding of cultural norms and Ho language. Senior Adivasi colleagues in the analysis group brought many years of experience in qualitative research and community engagement.

## Measures to increase trustworthiness

Our overall approach to increase trustworthiness built upon the four principles summarised by Ahmed [29]: credibility achieved through extended involvement with concerned communities by Ekjut team members, discussions about personal biases and preconceptions throughout the research process, and triangulation of results between participant groups; a critical assessment of transferability drawing upon detailed descriptions of the communities' characteristics and clear description of the sampling process; dependability, which we achieved through an audit trail linked to the development of the Framework matrix; and confirmability, as established through peer debriefing. We report our study using the Consolidated Criteria for Reporting Qualitative Studies (COREQ) checklist (S2 Text) and our Inclusivity Questionnaire is included as S3 Text.

## Results

In total, we interviewed 88 participants: 26 adolescents aged 10–14 years, 27 adolescents aged 15–19 years, 13 teachers, 11 parents, three health workers, and eight Ekjut peer facilitators aged 18–25 years. All participants were from the rural Ho community, including teachers and health workers. Our key findings are mapped as Fig 1 (Overview of themes in qualitative data) and summarised below.

### Concepts of distress

The most commonly used term to describe distress among adolescents was "udu", a Ho root word used as a noun to express thought, rumination, worry or concern, or as part of a verb to express internal states, for example in "udu

**Fig 1. Overview of themes in qualitative data.**

ngtana" ("I feel worried") or "udu tanang" ("I am thinking/reflecting"). "Udu" could also be further modulated to describe deep unhappiness, as in "Essu purre uddu raing taikena." ("I was deeply sad and distressed"). A boy aged 15–19 years, explained:

"When we are scolded without reason, we feel extremely sad, [we] feel worried and remain worried the whole day."

Ho: "Immin purre udui tana - chimiten ayein sama sama te gonde arang ni ko emetein aanj esu duku udu gein adaya, enamein te esu pareshan ge atkareya ondo enamein te aanj meesingi udu reinj taina".

While "udu" was the most used term to describe distress, because of its link to "thought" more generally, it could also be used to mean "to remember" or "to memorise." For example, Suresh, a peer counsellor, often encouraged his adolescent group by saying, "Issue badiye appe pe udu keda" (It is very good that you remembered it [what was discussed before]"). The meaning of "udu" was therefore situational and depended on its linguistic modulation.

Other terms connected to "udu" included "purre mon duku" and "ka sukuinjh tana", which were used to describe experiences of intense sadness, crying, emotional numbness, silence, loss of interest, self-blame, and thoughts of self-harm. Another boy aged 15–19 years shared:

"I used to smoke bidi [hand-rolled cigarettes]. Then I was made to understand the consequences, and I stopped smoking. Still, one accuses me of smoking. I became very sad and stressed, and I started crying."

Ho: "Anjh ayar bidi sibe taikinjh, bidi reya karab geya mein te ko somjao kidiniko ondo anjh bidi sib bagekidanj, enjh reyo een samayare essu purre en udu wue oondo dukku eeng, ondo aanyen raah aya."

Another term, "kurkur", was used to describe anger, often associated with feelings of threat, restriction, aggressiveness, irritability, isolation, refusing food and "talking back". Adolescents and parents said that those with distress commonly expressed anger, became more withdrawn, increased their consumption of addictive substances, and sometimes responded to parents or elders with defiance. In Ho, this was referred to as "kaji hureng", or "talking back". One parent explained:

"After our children experience problems and distress, their behaviour towards us changes. Yes, they get angry, become stubborn, "Anna sanag" (my wish), and later, talk back at us. They don't speak openly and seem to do whatever they want."

The terms "ka odong tana" and "giyu" described shyness and wanting to isolate from others. This often involved withdrawal, silence, covering the face, avoiding eye contact, and isolating oneself, as summarised by the phrase "hapa hapa taintana". Similarly, "boro", loosely translated as fear, was described as wanting to be alone, skipping meals, persistent worry, headaches, avoiding people and conversations, silence, and losing interest in previously enjoyable activities.

In some cases, emotional distress also manifested through symptoms such as loss of appetite, weight loss, and difficulty sleeping. A 19-year-old girl explained that she experienced such symptoms after losing her mother and her father's remarriage:

"Though we [she and her sister] were living, that living was equal to death, living like a dead body. There was no strength in the body, no energy, and we were neither eating nor drinking anything. Whether it was midnight or 1 o'clock, our tears never stopped."

Ho: "Jid do jide kana goye kan barabari paiti tanain ekdum goya kan ho lika main do na jana jum dai tana chi noon dai tana."

## Causes of distress

**Family responsibilities, gendered expectations, and their repercussions.**  Most adolescents described a growing burden of responsibilities as they got older. Both boys and girls reported having less time to play and increasing expectations to contribute to the household. These expectations were gendered and had distinct consequences. Boys described pressure to earn money and contribute financially. Girls reported greater pressure to do household chores and less support for continuing their education, sometimes leading to them dropping out of school.

> "I feel restricted from playing, have pressure to wear appropriate clothes, and am asked to do household chores. We help with household chores. This makes me sad." (Girl, 10–14 years)

Reaching puberty marked a significant shift, especially for girls, as gendered expectations intensified. Some reported being discouraged from pursuing education:

> "Some of our parents stop us from going to school, make us work in the agricultural fields, and say, 'When you're grown up, you'll have to cook on mud stoves' (Hindi: "chullah hi to phuk na hai") - what's the need for studies?" (Girl, 10–14 years)

Teachers confirmed adolescents' accounts, noting that economic pressures and a lack of parental support made it difficult for many, especially girls, to stay in school.

> "Stress increases mostly when students reach 8th or 9th standard. They begin to support the family income and help their parents in the agricultural fields. Parents don't understand the importance of education. We teach to the best of our abilities, but the home environment doesn't support learning. Most parents are not educated and cannot guide their children. Some children don't have a caregiver at all. If she or he doesn't earn, then what will they eat?" (Secondary school teacher)

> "In Adivasi communities, we generally don't impose restrictions on our daughters and sons or discriminate. But gender preference in education is still happening, and it forces girls to drop out, even bright ones. They get less time for studies and eventually drop out of school." (Middle school teacher)

Older adolescents (15–19 years) spoke of distress linked to romantic relationships, early marriage, and blocked educational or employment aspirations. For married adolescents, these challenges intensified as they moved into their in-laws' homes and faced new emotional and physical burdens. One married adolescent aged 15–19 years shared:

> "Responsibilities and work burden increase for married adolescents: both of us were forced to work to earn money. Both of us are socially boycotted because of an inter-caste marriage. We faced a conflict that arose from both the community and family to live together because of an inter-caste marriage."

**Violence in the family, school, and community.**  Adolescents described violence - particularly in the home - as a key source of distress. Alcohol-related conflict between parents was commonly mentioned. Girls also feared harassment on the way to school, which in some cases led to dropping out, especially where violence had occurred or was rumoured in the community, as described here by both younger and older girls.

"While going to school, we experienced eve-teasing, molestation, and comments passed by boys. Because of this, we stopped going to school. There is also a strong fear because some girls have been rumoured to be murdered." (Girl, 10–14 years)

In a group discussion featuring a vignette where a girl called Moni stopped going to school, girls aged 15–19 years said:

"Moni may have been facing harassment in school, that's why she stopped going to school. Moni thinks that she is growing up and restricting herself from playing. She must have faced violence, which she was not able to share with her parents and friends, that's why she is silent."

When the same vignette was discussed with boys aged 15–19 years, they said:

"Family members must have said something. Her desires are not fulfilled. She had not been sent to school; she was restricted from meeting friends. When we are stressed, we do not get a sense of time and do not know what to do. She must have been scolded when she returned home from outside. She must have been scolded by her parents when she was having food, that's why she stopped eating food."

Boys aged 10–14 also reported experiencing violence, including bullying at school, in the community, and at home.

**Family separation or loss.** Teachers and parents described how families' economic hardship led some adolescents to be sent to work outside the village.

"Nowadays, adolescents, especially girls, migrate due to compulsion, hunger, and other familial vulnerabilities. Then COVID worsened the situation. They stayed out of school for a long time and never resumed afterwards." (Middle school teacher)

"Some of us send our children by (falsely) revising their age on their Aadhaar ID [Identity Card]. We could have stopped it, but due to our family's financial situation, we send them." (Parent)

Some adolescents spoke about the emotional toll of separation, remarriage, or the loss of a parent.

"After my father's second marriage, we two sisters were in depression. From morning to night, we cried and supported each other." (Girl, 15-19 years)

"After my father died, I cried for many days. I would see my father whenever anyone came in front of me, and whenever someone in my village died, I felt it was my father who had died again." (Girl, 10-14 years)

## Social impacts of distress

Distress often manifested as social withdrawal, creating a vicious cycle of isolation, limitations in daily activities, and emotional strain. A high school teacher observed:

"During the agricultural season, if a child is absent for 2–3 days, there is a fear they will not return to class. When adolescents get married or face problems, many do not share their issues; they become silent, hide their feelings, and cannot concentrate on studies. After some time, they drop out. Some adolescents do share, but the majority do not."

Parents also expressed concern about their children growing emotional distant and the lack of safe spaces to talk:

> "Due to changes in adolescence, our children do not share or talk with us about their feelings, like happiness or sadness. Whenever they feel pain, they never share it with us. Wherever they feel like going, they go [without asking]." (Parent)

Distress also contributed to impulsive decisions to leave home, marry early or migrate for work as strategies to escape emotional pain or conflict:

> "Some adolescents lose their temper easily. If parents say something, they react negatively. They leave the house and start working outside. At a young age, they leave home and migrate for work." (Teacher)

Distressing thoughts also sometimes escalated into extreme ideas of self-harm or suicide, especially among younger adolescent girls.

> "If an adolescent girl goes through situations like Moni's, some may attempt suicide, and some may elope with a boy at an early age." (Girl, 15–19 years)

> "When I asked for money (pocket money), I was denied and scolded. While crying, I requested the money again because I needed it badly. At that time, I felt a great sense of anger and thought about drowning myself. When we are angry, we can go to any extent, even to suicide, because we cannot handle the anger." (Girl, 10–14 years)

> "Girls who are often scolded think about leaving the house or even about dying." (Girl, 10–14 years)

These thoughts were often linked to family restrictions, lack of emotional support, and feeling powerless or unheard.
Early marriage and substance use were perceived as both causes and consequences of distress. An adolescent boy aged 15–19 years said:

> "When the aspiration is not fulfilled, according to them, they start taking addictive products It is influenced by peers. Some take it with happiness, and some when they are under stress."

Similarly, during a group discussion, a girl aged 15–19 years said:

> "Seeing their parents and elders, some adolescents take addictive products, which affects their health. They become thin and lose interest in studies."

### Help-seeking preferences

When discussing access to support - whether formal mental health services or informal help - most adolescents cited confidentiality concerns and lack of family support as major barriers. Judgmental attitudes from parents could also discourage them from seeking help:

> "We are called *bekar* [Hindi for useless] and taunted at home. This makes us feel bad and less willing to talk about our problems." (Boy, 10–14 years)

Adolescents identified siblings, similarly aged in-laws, and friends as their main sources of support when they experienced distress. One Ekjut youth facilitator shared a poignant Ho-language poem describing a sister's words to her brother; the

poem both highlighted the strength of bonds between siblings and the challenges a girl may face when being separated from her family after marriage:

> "Dear brother, when a girl is born - be it a sister or a daughter -
>
> Like a ripe fruit of a vine, she is like a "Siyali" [creeper plant native to South Asia] seed ready to be sown elsewhere.
>
> One day, after marriage, I too will leave for another village or city.
>
> Perhaps, I won't be able to come to my father's funeral,
>
> Nor will I be able to see my mother one last time.
>
> Brother, please send me a message or send me a letter.
>
> When I hear the news through a messenger,
>
> Or read in a letter about the passing of our parents,
>
> I will raise my hands, beat my chest,
>
> My eyes will fill with tears, sorrow will overwhelm me,
>
> And the tears will keep falling.
>
> Even if I run, I won't reach in time for the final rituals.
>
> Even if I walk, I won't make it.
>
> I'll cover myself with a cloth in the middle of the courtyard,
>
> While on the riverbank, the funeral pyre will burn in silence."

Adolescents who had experience of support from community health workers such as Accredited Social Health Activists, or from older, trained peers, valued these greatly. Group meetings on adolescent health led by local youth facilitators trained by Ekjut were often viewed as safe and supportive spaces.

> "In those meetings, people speak kindly, share information, and listen. That makes us want to talk." (Girl, 15–19 years)

Adolescents said they were more likely to share their problems when someone praised their strengths, spoke to them kindly, taught them something new, and kept conversations confidential. Preferred locations to discuss distress and strategies to get better included "panchayat bhawans" (community centres), "chabutaras" (raised village platforms), or other quiet places where conversations would not be disturbed. While most adolescents preferred group settings, some older boys expressed a preference for one-to-one conversations for personal topics.

## Discussion

This is the first qualitative study on adolescent mental health from a large Indigenous Adivasi community in India. We identified several terms for distress, with "udu" the most significant. Distress was rooted in gendered expectations and increasing family responsibilities, alongside violence at home, bullying, and street harassment of girls. Separation due to migration, remarriage, or parental death was also important. Social impacts included isolation, strained relationships, early marriage, work migration, and sometimes suicidal thoughts or attempts. Adolescents mainly sought support from siblings,

trusted teachers, health workers, and experienced older peers. These were viewed as acceptable sources of help both now and in the future.

Our study is one of only a handful to explore "emic" Adivasi concepts of distress [30,31]. The term "udu" bears some similarities to the concept of stress or "tension," an idiom recognised in other South Asian contexts. Like "tension", "udu" is not equivalent to psychiatric categories and can take on different meanings depending on context [32]. Adolescents used "udu" to describe stress, but those described as having strong "udu" had symptoms such as low mood, loss of interest in normal activities, irritability, repetitive thoughts, reduced appetite and sleep, and somatic complaints. Unlike "tension" however, "udu" does not appear to rely on a "hydraulic" ethno-psychological model in which emotional pressure builds and is released through physical or affective symptoms. Instead, "udu" seems more closely aligned with idioms centred on thought, such as "thinking too much", a globally prevalent idiom of distress in which repetitive thinking (rumination, worries) is a driver of emotional pressure [33]. Rather than pointing to a single psychological theory of adolescent distress or depression, our findings highlight the importance of models that consider both negative interpersonal dynamics, especially in the family, and broader socio-cultural factors that increase exposure to adverse life events like school dropout, financial hardship, or separation from parents [34].

What do our findings add to knowledge on sources of distress in Adivasi communities? Past reviews of quantitative Indian studies with rural adolescents, including those from Adivasi communities, identified strained family relationships, academic challenges, and economic hardship as risk factors for depression and anxiety [35,36]. Our findings lend support to the broad patterns identified in these studies but also suggest that gender-related expectations, violence, and discrimination are major contributors to distress, in line with limited existing Adivasi-specific research [37]. Our study also highlights key inflection points in early and later adolescence when gender, age, and family circumstances intersect to heighten vulnerability. For example, young girls reaching puberty were often expected to take on more household responsibilities, sometimes at the expense of their education, and reported increased exposure to harassment. Similarly, both boys and girls aged 15–19 years faced pressure to contribute financially or prepare for marriage. Overall, our study underscores the critical role of social determinants in shaping adolescent mental health. Adolescents and teachers described how economic hardship impacted caregivers' emotional availability and increased pressure on adolescents to work, compounding their distress. These insights suggest the need for upstream, preventive interventions, including livelihood and social protection support for families, consistent with broader frameworks addressing the social determinants of mental health [4].

A key strength of this study was its grounding in a community mental health programme developed as part of Ekjut's 20+year partnership with Ho communities. Following the principle of "no survey without service," Ekjut was able to provide psychosocial counselling support and referrals to participants who needed help. Peer interviewers from the community who were fluent in Ho and familiar with local structures helped refine topic guides, conducted interviews, and contributed to data analysis. Interviews with adolescents were conducted in Ho, improving rapport and reducing the risk of meaning being lost in translation. Our study also had limitations. We included fewer younger adolescents (10–14 years) than older ones, as young girls were more reluctant to participate. Additionally, our vignette-based approach, while helpful for discussing distress, was still somewhat complex for younger participants, who might have responded better to simpler questions. We also did not identify any adolescents with non-binary gender identities, though this may reflect discomfort with disclosure rather than absence, given adolescence is often a time of identity exploration. Finally, while our methodological approach is transferable, our findings are context specific. The concepts of distress and patterns of help-seeking we identified may not be generalisable to all Adivasi communities in India given their diversity.

What do our findings mean for future action? While many sources of distress we identified overlapped with those identified in past Indian studies, differences in socio-economic conditions and access to school in Adivasi communities mean that some of their help-seeking preferences may be distinct to those in urban or southern Indian contexts described in past reviews [34]. Limited access to smartphones and lower levels of school enrolment in secondary school, particularly

for girls, suggest that focusing mainly on developing digital and school-based mental health interventions may inadvertently exclude the most underserved. Mental health programmes could also build on the high esteem afforded to youth leaders in many Adivasi communities, where young people serving as peer educators or community health workers are highly valued. Evidence from India and other contexts suggests that community-based mental health interventions delivered by trained older peers or lay counsellors can be effective to lower symptoms of anxiety and depression in adolescents [38,39]. Such strategies may be particularly promising in Adivasi areas, given respect for youth leadership and the considerable workload of existing community health workers. In addition, our findings suggest that community-based mental health programmes for adolescents must be linked with wider efforts to reduce the socio-economic adversity faced by their families, including by improving access to government entitlements and connecting families with livelihood opportunities. Efforts to combine preventive psychological interventions delivered through task-shifting with actions on the social determinants of mental health have already shown promise to reduce the incidence of depression in children and could be further expanded [40]. Without action on these social determinants of mental health, psychological interventions may have limited acceptability and medium- to long-term effectiveness.

## Conclusion

Our study identified several key terms to describe distress among adolescents in Ho Adivasi communities of Jharkhand, documented causes and impacts of distress, as well as preference for help-seeking. This and other research highlight the need for community-based mental health promotion and treatment support for adolescents in underserved Adivasi communities. In addition, our work suggests that any attempt to improve mental health among young people should address livelihood support and social protection for families, in addition to providing psychological therapy and appropriate referral mechanisms for adolescents.

## Supporting information

**S1 Text. Topic guides in English and Hindi.**
(DOCX)

**S2 Text. Consolidated Criteria for Reporting Qualitative Studies (COREQ).**
(DOCX)

**S3 Text. Inclusivity questionnaire.**
(DOCX)

**S1 Table. Details of qualitative methods used for each participant group.**
(DOCX)

**S1 Data. Final coding matrix.**
(XLSX)

## Author contributions

**Conceptualization:** Subhashree Samal, S. Prathyush, Sumitra Gagrai, Nirmala Nair, Sachin Barbde, Audrey Prost.

**Data curation:** Subhashree Samal, Nibha Das, Savitri Banra, Suresh Purty, Madhuri Purty, Birsingh Mundri, Dumbi Bodra, Mati Murmu, Sunita Kalundia, Yadav Leyangi.

**Formal analysis:** Subhashree Samal, S. Prathyush, Sumitra Gagrai, Nibha Das, Nitima Lamay, Savitri Banra, Suresh Purty, Madhuri Purty, Birsingh Mundri, Dumbi Bodra, Mati Murmu, Sunita Kalundia, Yadav Leyangi, Sachin Barbde, Audrey Prost.

**Funding acquisition:** Andrea Danese, Kelly Rose-Clarke, Abhijit Nadkarni, Sachin Barbde, Audrey Prost.

**Investigation:** Subhashree Samal, S. Prathyush, Sumitra Gagrai, Nibha Das, Nitima Lamay, Mati Murmu, Sunita Kalundia, Yadav Leyangi, Sachin Barbde, Audrey Prost.

**Methodology:** Subhashree Samal, S. Prathyush, Sumitra Gagrai, Nibha Das, Nitima Lamay, Savitri Banra, Suresh Purty, Madhuri Purty, Birsingh Mundri, Dumbi Bodra, Mati Murmu, Sachin Barbde, Audrey Prost.

**Project administration:** Subhashree Samal, S. Prathyush, Sachin Barbde, Audrey Prost.

**Resources:** Sachin Barbde.

**Supervision:** Subhashree Samal, S. Prathyush, Sumitra Gagrai, Nibha Das, Nitima Lamay, Mati Murmu, Nirmala Nair, Prasanta Tripathy, Sachin Barbde, Audrey Prost.

**Visualization:** Subhashree Samal.

**Writing – original draft:** Subhashree Samal.

**Writing – review & editing:** Subhashree Samal, S. Prathyush, Sumitra Gagrai, Nibha Das, Nitima Lamay, Savitri Banra, Suresh Purty, Madhuri Purty, Birsingh Mundri, Dumbi Bodra, Mati Murmu, Sunita Kalundia, Yadav Leyangi, Nirmala Nair, Prasanta Tripathy, Patrick Smith, Andrea Danese, Kelly Rose-Clarke, Abhijit Nadkarni, Urvita Bhatia, Saicha Naik Bandiwadekar, Brinda Singh Raikwar, Devika Gupta, Sachin Barbde, Audrey Prost.

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
