## [Decision Letter · Decision Letter 0]

18 Jan 2026

PMEN-D-25-00421

Concepts of psychosocial distress and help-seeking preferences among Indigenous adolescents: a qualitative study from Jharkhand, India

PLOS Mental Health

Dear Dr. Prost,

Thank you for submitting your manuscript to PLOS Mental Health and I am very sorry for the severe delay. After careful consideration of the reviewer reports, which we have now received, we feel that your paper has merit but does not fully meet PLOS Mental Health’s publication criteria as it currently stands. Therefore, we invite you to submit a revised version of the manuscript that addresses the points raised during the review process.

Please address all of the comments raised, which you can find below and attached.

We look forward to receiving your revised manuscript.

Kind regards,

Karli Montague-Cardoso

Staff Editor

PLOS Mental Health

Journal Requirements:

1. Please include a complete copy of PLOS’ questionnaire on inclusivity in global research in your revised manuscript. Our policy for research in this area aims to improve transparency in the reporting of research performed outside of researchers’ own country or community. The policy applies to researchers who have travelled to a different country to conduct research, research with Indigenous populations or their lands, and research on cultural artefacts. The questionnaire can also be requested at the journal’s discretion for any other submissions, even if these conditions are not met.  Please find more information on the policy and a link to download a blank copy of the questionnaire here: https://journals.plos.org/mentalhealth/s/best-practices-in-research-reporting. Please upload a completed version of your questionnaire as Supporting Information when you resubmit your manuscript.

i. Please clarify all sources of financial support for your study. List the grants, grant numbers, and organizations that funded your study, including funding received from your institution. Please note that suppliers of material support, including research materials, should be recognized in the Acknowledgements section rather than in the Financial Disclosure.

ii. State the initials, alongside each funding source, of each author to receive each grant. For example: "This work was supported by the National Institutes of Health (####### to AM; ###### to CJ) and the National Science Foundation (###### to AM)."

iii. State what role the funders took in the study. If the funders had no role in your study, please state: “The funders had no role in study design, data collection and analysis, decision to publish, or preparation of the manuscript.”

iv. If any authors received a salary from any of your funders, please state which authors and which funders.

3. Please ensure that your Ethics Statement is available in its entirety at the beginning of your Methods section, under a subheading 'Ethics Statement'.

4. We have noticed that you have uploaded Supporting Information files, but you have not included a list of legends. Please add a full list of legends for your Supporting Information files after the references list.

5. We note that you have indicated that there are restrictions to data sharing for this study. For studies involving human research participant data or other sensitive data, we encourage authors to share de-identified or anonymized data. However, when data cannot be publicly shared for ethical reasons, we allow authors to make their data sets available upon request. For information on unacceptable data access restrictions, please see http://journals.plos.org/plosone/s/data-availability#loc-unacceptable-data-access-restrictions.

Additional Editor Comments (if provided):

Reviewers' comments:

Reviewer's Responses to Questions

**Comments to the Author**

1. Does this manuscript meet PLOS Mental Health’s publication criteria? Is the manuscript technically sound, and do the data support the conclusions? The manuscript must describe methodologically and ethically rigorous research with conclusions that are appropriately drawn based on the data presented.? Is the manuscript technically sound, and do the data support the conclusions? The manuscript must describe methodologically and ethically rigorous research with conclusions that are appropriately drawn based on the data presented.

Reviewer #1: Yes

Reviewer #2: Yes

2. Has the statistical analysis been performed appropriately and rigorously?

Reviewer #1: N/A

Reviewer #2: Yes

3. Have the authors made all data underlying the findings in their manuscript fully available (please refer to the Data Availability Statement at the start of the manuscript PDF file)?

The PLOS Data policy requires authors to make all data underlying the findings described in their manuscript fully available without restriction, with rare exception. The data should be provided as part of the manuscript or its supporting information, or deposited to a public repository. For example, in addition to summary statistics, the data points behind means, medians and variance measures should be available. If there are restrictions on publicly sharing data—e.g. participant privacy or use of data from a third party—those must be specified.requires authors to make all data underlying the findings described in their manuscript fully available without restriction, with rare exception. The data should be provided as part of the manuscript or its supporting information, or deposited to a public repository. For example, in addition to summary statistics, the data points behind means, medians and variance measures should be available. If there are restrictions on publicly sharing data—e.g. participant privacy or use of data from a third party—those must be specified.

Reviewer #1: Yes

Reviewer #2: Yes

4. Is the manuscript presented in an intelligible fashion and written in standard English?

Reviewer #1: Yes

Reviewer #2: Yes

5. Review Comments to the Author

Reviewer #1: Dear Author

Major

1. Along with positionality, the authors could summarise the measures taken for rigor and trustworthiness.

2. Consider using COREQ checklist while reporting, though most of the points are covered, some aspects are to be addressed.

3. Characteristics of participants could be summarized in the beginning of results

4. The authors mention that they use framework matrix analysis but I could not see it in the results. It looked like only a thematic analysis. To justify or clarify how the analysis fits the framework matrix analysis. I could not see multi-case comparisons.

5. Add strengths and limitation of the study.

Minor

1. Line 71: “c.9%” there is a typing error

2. The methodological orientation can come before the study participants.

Reviewer #2: This manuscript presents a rigorous and richly contextualised qualitative study exploring Indigenous Adivasi adolescents’ concepts of psychosocial distress and help-seeking preferences in rural Jharkhand, India. The study addresses an important and under-researched population and makes a valuable contribution to the literature on idioms of distress, adolescent mental health, and community-based care in low-resource Indigenous settings. The methodological approach is generally strong, and the findings are well supported by data. However, some sections would benefit from greater analytic tightening and clearer articulation of the paper’s conceptual and practical contributions.

1. While the richness of the qualitative data is a strength, the Results section would benefit from greater analytic condensation. Several quotations illustrate similar points, and selective reduction of excerpts particularly where meanings overlap could improve clarity and strengthen the analytic narrative without loss of nuance.

2. The concept of “udu” emerges as a central idiom of distress in this study. However, the Discussion could further strengthen the manuscript by more explicitly articulating how “udu” advances existing idioms-of-distress literature (e.g., in relation to “tension” or “thinking too much”) and what this implies for assessment, engagement, or intervention design in adolescent mental health programmes. The manuscript would benefit from a more focused discussion of how the findings could inform the design of culturally grounded, community-based adolescent mental health interventions, particularly in Indigenous or rural contexts where access to formal services is limited.

3. Framework analysis is appropriate, but some procedural details are brief. The authors may consider adding brief clarification on how coding disagreements were resolved and how saturation was operationalised in practice, to further strengthen transparency and methodological rigor.

6. PLOS authors have the option to publish the peer review history of their article (what does this mean?). If published, this will include your full peer review and any attached files.). If published, this will include your full peer review and any attached files.

**Do you want your identity to be public for this peer review?** For information about this choice, including consent withdrawal, please see our Privacy Policy..

Reviewer #1: **Yes:** Mahalakshmy ThulasingamMahalakshmy Thulasingam

Reviewer #2: No

Figure Resubmissions:

---

## [Editor Report · Decision Letter 1]

27 Mar 2026

Concepts of psychosocial distress and help-seeking preferences among Indigenous adolescents: a qualitative study from Jharkhand, India

PMEN-D-25-00421R1

Dear Professor Prost,

We are pleased to inform you that your manuscript 'Concepts of psychosocial distress and help-seeking preferences among Indigenous adolescents: a qualitative study from Jharkhand, India' has been provisionally accepted for publication in PLOS Mental Health.

Best regards,

Karli Montague-Cardoso

Staff Editor

PLOS Mental Health